# Frontline Healthcare Workers’ Knowledge and Perception of COVID-19, and Willingness to Work during the Pandemic in Nepal

**DOI:** 10.3390/healthcare8040554

**Published:** 2020-12-11

**Authors:** Dipak Prasad Upadhyaya, Rajan Paudel, Dilaram Acharya, Kaveh Khoshnood, Kwan Lee, Ji-Hyuk Park, Seok-Ju Yoo, Archana Shrestha, Bom BC, Sabin Bhandari, Ramgyan Yadav, Ashish Timalsina, Chetan Nidhi Wagle, Brij Kumar Das, Ramesh Kunwar, Binaya Chalise, Deepak Raj Bhatta, Mukesh Adhikari

**Affiliations:** 1School of Medicine, Case Western Reserve University, Euclid Ave, Cleveland, OH 44106, USA; 2Central Department of Public Health, Institute of Medicine, Tribhuvan University, Kathmandu 44600, Nepal; paudel.rajan@gmail.com; 3Department of Preventive Medicine, College of Medicine, Dongguk University, Gyeongju 38066, Korea; dilaramacharya123@gmail.com (D.A.); kwaniya@dongguk.ac.kr (K.L.); skeyd@naver.com (J.-H.P.); 4Department of Epidemiology of Microbial Diseases, Yale School of Public Health, Yale University, New Haven, CT 06510, USA; kaveh.khoshnood@yale.edu; 5Department of Public Health, School of Medical Sciences, Kathmandu University, Dhulikhel 45200, Nepal; archana@kusms.edu.np; 6Institute for Implementation Science and Health, Kathmandu 44600, Nepal; 7Ministry of Health and Population, Kathmandu 44600, Nepal; bombc2011@gmail.com (B.B.); ramgderma@gmail.com (R.Y.); ashish.timalsina@gmail.com (A.T.); uchetan@hotmail.com (C.N.W.); dasbrijkumar@gmail.com (B.K.D.); ramesh.kunwar04@gmail.com (R.K.); deepakbhattaph@gmail.com (D.R.B.); mukesh.adhikari@yale.edu (M.A.); 8B.P Koirala Institute of Health Sciences, Dharan 56700, Nepal; sabin7000@gmail.com; 9Graduate School for International Development and Cooperation, Hiroshima University, Hiroshima 739-8527, Japan; binayachalise@gmail.com; 10Department of Health Policy and Management, Yale School of Public Health, Yale University, New Haven, CT 06510, USA

**Keywords:** COVID-19, frontline healthcare workers, knowledge, perception, willingness, Nepal

## Abstract

This study investigated the contextual factors associated with the knowledge, perceptions, and the willingness of frontline healthcare workers (FHWs) to work during the COVID-19 pandemic in Nepal among a total of 1051 FHWs. Multivariable logistic regression analysis was applied to identify independent associations between predictors and outcome variables. Of the total study subjects, 17.2% reported inadequate knowledge on COVID-19, 63.6% reported that they perceived the government response as unsatisfactory, and 35.9% showed an unwillingness to work during the pandemic. Our analyses demonstrated that FHWs at local public health facilities, pharmacists, Ayurvedic health workers (HWs), and those with chronic diseases were less likely, and male FHWs were more likely, to have adequate knowledge of COVID-19. Likewise, nurses/midwives, public health workers, FHWs from Karnali and Far-West provinces, and those who had adequate knowledge of COVID-19 were more likely to have satisfactory perceptions towards the government response. Further, FHWs—paramedics, nurse/midwives, public health workers, laboratory workers—FHWs from Karnali Province and Far-West Province, and those with satisfactory perceptions of government responses to COVID-19 were predictors of willingness to work during the COVID-19 pandemic. These results suggest that prompt actions are required to improve FHWs’ knowledge of COVID-19, address negative perceptions of government responses, and motivate them through specific measures to provide healthcare services during the pandemic.

## 1. Introduction

In December 2019, an outbreak of a pneumonia-like illness was first detected in Wuhan, Hubei Province of China [1], and subsequently, faced with an escalating number of cases beyond China, the World Health Organization (WHO) declared the outbreak a pandemic [2]. Based on available evidence, the infection is transmitted between individuals via nasopharyngeal droplets or saliva [3]. Furthermore, no vaccine or effective treatment for COVID-19 is currently available [3].

Nepal is a small country in South Asia that shares a border with China and observed its first case of COVID-19 on 25 January 2020 [4]. In the first half of May 2020, Nepal experienced an explosive increase in cases; more than three-fourths of all cases recorded to date occurred during this period. As of 4 October 2020, Nepal reported 132,246 cases and 739 deaths [5]. Concern has been expressed that health systems in low-income countries such as Nepal are not sufficiently resilient to tackle a crisis presented by COVID-19. Due to resource constraints and a weak health system structure, rapid diagnosis of suspected cases and contact tracing are challenging [6]. Studies have shown that knowledge of infectious diseases is greatest among doctors and nurses [7,8]. In addition, age, sex, educational status, and pre-existing medical conditions have been shown to affect health workers’ (HWs) knowledge of Middle East Respiratory Syndrome (MERS) and Severe Acute Respiratory Syndrome (SARS) [9,10].

The primary sources of information about COVID 19 are from international health organizations such as the Center for Disease Control (CDC), the WHO and Ministry of Health, and social media. Moreover, the effectiveness of healthcare sectors during public health emergencies primarily depends on the availability, motivation, and skills of frontline healthcare workers (FHWs), and thus knowledge and their perceived willingness to work during uncertain times is essential [11], because appropriate perceptions and willingness to work during a pandemic are prerequisites of HW motivation to provide necessary treatment and to take the preventive actions required for the impact of a pandemic. Studies have shown that factors such are perceived personal risks, availability of personal protective equipment, family care obligations, HW gender, type of employment, personal confidence, defined role, dissemination of timely information, appropriate training, and personal health problems influence perceptions and willingness to work during pandemics [11,12,13,14]. In the present study, FHWs were defined as doctors, nurses, paramedics, laboratory workers, pharmacists, pathologists, and public health workers who were directly involved in COVID-19 prevention and control and who have direct contact with confirmed or suspected cases during patient intake, screening, inspection, testing, transport, treatment, nursing, specimen collection, or pathogen detection.

The present study explored factors associated with the knowledge of COVID-19 of FHWs, their reactions to government interventions, and, most importantly, their perceived willingness to work during the pandemic. This study could also provide valuable and actionable information to Nepalese policymakers to allow the judicious allocation of scarce resources in the short run. In the long term, this study might guide those developing policies and programs. That might be instrumental in ensuring preparedness to meet the challenges posed by similar crises. Given this background, we aimed to investigate the contextual factors associated with the perceptions and willingness to work among Nepalese frontline healthcare workers during the COVID-19 pandemic to improve the prevention and management of future similar outbreaks.

## 2. Materials and Methods

### 2.1. Study Design, Participants, and Sampling

We conducted a nationwide web-based cross-sectional study using an online questionnaire between 1 May and 10 June 2020 among FHWs in Nepal in accord with the Checklist for Reporting Results of Internet Surveys (CHERRIES) [15]. All participating FHWs were aged 18 to 60 years old and ranged from high-level officials of the Ministry of Health and Population to paramedics working at the grassroots level in all seven provinces of Nepal, and excluded those subjects who were had mentally ill or unwilling to participate in the study. The research questionnaire was distributed to FHWs using the health workers’ network. Firstly, we had a medical doctor or a public health professional in each of the seven provinces to act as a coordinator and co-investigator in the team. Secondly, these seven coordinators sent FHWs they knew a link of the questionnaire and asked that these individuals send a Google link to other FHWs they knew. Fischer’s arctanh transformation as a power of 90% and a minimum correlation of 0.1 showed that the minimum sample size required was *n* = 1046. 

### 2.2. Survey Instrument and Data Collection

The online questionnaire included 33 questions on socioeconomic characteristics, HWs’ knowledge of COVID-19, perception toward government response to COVID-19, and perceived willingness to work during the pandemic. The responses were rated using a 5-point Likert Scale (“Strongly Agree”, “Agree”, “Neutral”, “Disagree” to “Strongly Disagree”). The sociodemographic characteristics investigated included age, gender, caste/ethnicity, and marital status. This section of the questionnaire also included questions about chronic diseases of HWs, their caretaking responsibilities for dependent family members, nature of their employment, and type of health facility at which they worked. There were three outcomes variables—knowledge of COVID-19, perception of government response in the management of COVID-19, and willingness to work during the COVID-19 pandemic. Knowledge of COVID-19 of the subjects was assessed using 20 structured questions that included information about COVID-19 such as agent, host and environmental factors, modes of transmission, diagnosis, and preventive and control measures. Likewise, perception of government response in the management of COVID-19 consisted of 25 structured questions that contained the information-timeliness of information updates from health authorities, whether there was shortage of personal protective equipment and other logistics, effectiveness of existing government response to the COVID-19 pandemic, and adequate support from government administrative bodies. Similarly, we had 35 structured questions to measure the willingness of FHWs to work during the pandemic that included information relating to self-infection, time factors, rationality, training, work place, work environment and liberty given to work during the COVID-19 pandemic. We dichotomized the health workers’ knowledge of COVID-19 (total score of 20, one score per question) into “adequate knowledge” for those whose score were >12 of 20 knowledge-related questions (60%) and “inadequate knowledge” for those <12 (<60%). Similarly, health workers’ perception of government responses to COVID-19 prevention and control consisted of 25 questions for which those who scored >60% (>15 of 25) were categorized as having a “satisfactory perception” or “unsatisfactory perception <15”. Further, health workers’ willingness to work had a total of 35 questions that were categorized into “willing to work” >21 (>60%) and “unwilling to work” <21 score. Knowledge of COVID-19 was assessed based on knowledge of the causative agent, mode of transmission, proper use of Personal Protective Equipment (PPE), infection prevention measures, and public health impact of the pandemic. Reaction to government response was determined by assessing response effectiveness, timeliness of information provided, provision of supplies, support received from administrative staff, and elected representatives. Factors influencing willingness to work during the pandemic were risk of self-infection, healthcare service rationing, the requirement to work overtime, working with untrained HWs, deployment to another duty station, family risk, and ability to choose whether to work or not during the pandemic.

The questionnaire was prepared based on national COVID-19 guidelines issued by the Ministry of Health and Population of Nepal [16] and the World Health Organization resource center guidelines for HWs on COVID-19 [17]. A team of medical doctors, public health workers, and an academic assessed the questionnaire for validity and relevance. Before conducting the survey, we conducted a pilot study on 30 participants to assess the questionnaire items’ reliability. The analysis revealed an overall Cronbach’s alpha score of 0.77, indicating higher internal consistency [18]. The questionnaire was prepared as a Google Form, and Facebook Messenger was used to send the Google form link to participants [19,20]. Only two research team members had access to the data repository to maintain data confidentiality, stored on a password-protected computer.

### 2.3. Data Management and Statistical Analysis

The data collected were downloaded in the form of a spreadsheet and checked for duplications and technical errors. After confirming the completeness, we exported the data to R Studio Software (R studio, Boston, MA, USA, version 1.2.5042) for full analysis. Sociodemographic characteristics were subjected to descriptive analysis using the Table 1 package in R software, and results are presented as frequencies, percentages, or as means and standard deviations. Univariate logistic regression analysis was used to assess factors associated with adequate knowledge, satisfaction with the government response, and willingness to work using the *finalfit* package in R [21]. Parsimonious multivariate models were created for each dependent variable and included independent variables found to be significant (*p*-value < 0.05) by univariate analysis. The data did not show the overly inflated outliers. We also checked for multicoliniariy before entering variables to multivariable logistic regression analysis. Coefficients in the regression models were transformed into odds ratios with 95% confidence intervals. *p*-values of <0.05 were considered significant. 

### 2.4. Ethics Statement

The study’s ethical approval was obtained from the Nepal Health Research Council (approval no: 329/2020 P). The first page of the questionnaire detailed the study objective, benefits, and harm. HWs provided e-consent before participating in the study. Participants were informed that they could leave the study at any time. Participation was voluntary and anonymous.

## 3. Results

### 3.1. Sociodemographic Characteristics

A total of 1051 FHWs participated in the study—725 (68%) were men, with a median age of 31 years. The majority (57.4%) were of Brahmin/Chhetri caste/ethnicity, married (64.4%) and only 35.3% were medical doctors. Province-wise study subjects ranged from 6.6% (lowest) in Gandaki Province to the highest (19.4%) being from Bagmati Province. Nearly 60% of FHWs were permanent government employees. More than 25% worked in local-level public health facilities, such as health posts, primary healthcare centers, community health units, and urban health centers; 13.5% of FHWs reported having a chronic disease; 64% of FHWs had family members of younger than five years old or more than 60 years old who needed their care and support (Table 1).

**Table 1 healthcare-08-00554-t001:** Sociodemographic characteristics of health workers in Nepal.

Sociodemographic Characteristics	Frequency (%)
	(*n* = 1051)
Age, in years, Median	31.0 (8)
Gender	
Female	326 (31.0)
Male	725 (68.0)
Caste/ethnicity	
Brahmin/Chhetri	603 (57.4)
Madhesi/Muslim	209 (19.9)
Janajati	174 (16.6)
Dalit	34 (3.2)
Other	31 (2.9)
Marital Status	
Married	677 (64.4)
Unmarried	374 (35.6)
Professional Category	
Doctor	371 (35.3)
Paramedics	308 (29.3)
Nurse/Midwife	173 (16.5)
Public Health Workers	122 (11.6)
Lab Worker	51 (4.9)
Other (Pharmacist, Ayurveda HW etc.)	26 (2.5)
Province	
Province 1	193 (18.4)
Province 2	156 (14.8)
Bagmati Province	204 (19.4)
Gandaki Province	69 (6.6)
Province 5	153 (14.6)
Karnali Province	111 (10.6)
Far-West Province	165 (15.7)
Type of Job	
Permanent	613 (58.3)
Temporary or Contract	438 (41.7)
Types of Health Facility	
Federal and Provincial managerial agencies ^Ø^	134 (12.7)
Teaching Hospital	191 (18.2)
Public Hospital	240 (22.8)
Private Hospital	132 (12.6)
Local public health facilities ^§^	292 (27.8)
Local-level managerial agencies ^⊗^	62 (5.9)
Presence of Chronic Disease	
No	909 (86.5)
Yes	142 (13.5)
HWs with caretaking responsibility for children less than 5 years or elderly more than 60 years	
No	381 (36.2)
Yes	670 (63.7)

^Ø^ Consists of the Ministry of Health and Population, the Department of Health Services, the Ministry of Social Development at the province level, Provincial health directorate, and health offices. ^§^ Consists of health posts, primary healthcare centers, community health units, and urban health centers at the local level. ^⊗^ Consists of metropolitan, sub-metropolitan, municipalities, and rural municipalities.

### 3.2. Health Workers’ Knowledge of COVID-19

Table 2 shows the frontline health workers’ knowledge and their associates. Of the total 1051 FHWs, 17.2% were found to have inadequate knowledge on COVID-19. The final adjusted multivariable logistic regression analyses demonstrated that FHWs at local public health facilities (AOR: 0.35; 95% CI: 0.17–0.68) were less likely to have adequate knowledge of COVID-19 compared to Federal and Provincial Managerial Agencies, as were other types of FHWs (pharmacists and Ayurvedic HWs) (AOR: 0.33; 95% CI: 0.14–0.80) compared to doctors. Those with chronic diseases (AOR: 0.58; 95% CI: 0.37–0.91) were less likely compared to those without, and male FHWs (AOR: 1.60; 95% CI: 1.02–2.47) were more likely to have an adequate knowledge of COVID-19.

### 3.3. Health Workers’ Reactions to Government Response to COVID-19 Pandemic

The majority of FHWs (63.6%) perceived the government response to COVID-19 management as unsatisfactory. Our adjusted multivariable logistic regression analysis revealed that nurses/midwives (AOR: 2.10; 95% CI: 1.38–3.18) and public health workers (AOR: 1.83; 95% CI: 1.07–3.11) were found to have higher odds of having a satisfactory perception towards the government response compared to doctors, as were FHWs from Karnali Province (AOR: 2.62; 95% CI: 1.52–4.53), and Far-West Province (AOR: 1.72; 95% CI: 1.06–2.80) compared to Bagmati Province, as well as those who had adequate knowledge of COVID-19 (AOR: 3.86; 95% CI: 2.51–6.16) (Table 3).

### 3.4. Health Workers’ Willingness to Work during the COVID-19 Pandemic

Table 4 shows that about 64% of FHWs reported a willingness to work under the challenging conditions created during the COVID-19 pandemic. Further, in accordance with the adjusted model, FHWs such as paramedics (AOR: 2.52; 95% CI: 1.79–3.58), nurses/midwives (AOR: 2.09; 95% CI: 1.40–3.17), public health workers (AOR: 2.40; 95% CI: 1.47–4.01), laboratory workers (AOR: 3.54; 95% CI: 1.77–7.61), were found to have increased willingness to work during the COVID-19 pandemic when compared to doctors, as were health workers from Karnali Province (AOR: 2.96; 95% CI: 1.62–5.64) and Far-West Province (AOR: 2.10; 95% CI: 1.28–3.48) compared to others. Those who perceived the government’s response to COVID-19 as satisfactory (AOR: 2.52; 95% CI: 1.79–3.58) were also found to have increased willingness to work compared to their counterparts.

## 4. Discussion

This is the first nationwide study on knowledge and perception of COVID-19 among FHWs, and their willingness to work during the COVID-19 pandemic situation in Nepal despite several other studies have been conducted in Nepal related to COVID-19. Approximately two in ten FHWs had inadequate knowledge of COVID-19, which is higher than that reported in a Chinese study, in which ≃11% demonstrated insufficient knowledge [22]. On the other hand, a study conducted by Bhagavathula et al. reported that 61% of health workers had poor knowledge about COVID-19 transmission [23]. These differences between rates may have been due to variations in the contents and criteria used for assessing the knowledge, and study subject attributes. Furthermore, the latter study was conducted in the first week of March 2020, and the Chinese study was conducted in the third week of May, when more information regarding COVID 19 was available and disseminated through different media.

Knowledge is crucial for establishing perceptions and preventive behaviors, affecting coping interventions to some degree [24]. We also found that male health workers were more likely to report adequate knowledge, which is consistent with the findings of another study conducted in Nepal [25]. This finding may be due to greater interaction and socialization by men, and gendered norms, which means men are more likely to overestimate, and women are likely to underestimate personal knowledge [26,27,28,29]. Our study also showed that pharmacists and Ayurvedic HWs had inadequate knowledge of COVID-19 as compared with doctors which is similar to a previous local area study conducted in Nepal [25]. This could obviously be a matter of content-specific higher educational achievement, and direct and higher levels of work exposure of medical doctors than those of pharmacists and Ayurvedic HWs. We found that FHWs in the local health facilities were less likely to have adequate knowledge than the FHWs in federal or provincial agencies. This could possibly be due to weaker implementation of COVID-19-related governmental interventions at the local level than at provincial or federal levels, and the provincial or federal level FHWs in Nepal are obviously more qualified than at local levels. In addition, FHWs with chronic diseases had inadequate knowledge of COVID-19, perhaps because of time limitations imposed by pre-existing conditions restricting studies on COVID-19. Therefore, these factors should be considered while disseminating available knowledge updating ongoing and the latest available information, education, and communication materials.

Our observation demonstrated that nearly two-thirds (63.6%) of FHWs perceived the government response to COVID-19 management as unsatisfactory. A slightly higher level of satisfaction with government response was reported in a survey conducted on the Nepalese general public in April 2020 (71.4%) [30]. The present study also showed that most FHWs (86%) experienced logistical shortcomings and reported inadequate supports from administrative (60%) and elected representatives (67.5%) (data not presented in the Table), which concurs with findings of a previous study [31]. Our observation additionally found that the professional category, province, type of health facility, and knowledge of COVID-19 were significantly associated with frontline health workers’ satisfaction with government response to the pandemic. Accordingly, nurses were found to be more likely to be satisfied with government response than frontline doctors. This perception difference might have been due to the differences in technical knowledge levels among doctors, nurses, and public health workers. Similarly, FHWs from Karnali and Far-West provinces were more likely to be satisfied with the government response than FHWs from Bagmati Province. However, the reasons for these provincial variations were not determined. In addition, FHWs from local public health facilities, teaching hospitals, and private hospitals had unsatisfactory perceptions compared to managerial level FHWs at the ministry level, which we attribute to different work experiences, as FHWs at health service outlets are directly exposed to risks and better understand the risks posed by logistical shortfalls than managerial-level FHWs. Interestingly, FHWs with adequate knowledge of COVID-19 had higher odds of satisfaction with government response than those with inadequate knowledge. Based on this evidence, prompt promotive actions such as provision of necessary medicaments, logistics, and empowerments in health workers’ knowledge are necessary from the Health Ministry of Nepal to create a favorable work environment during the pandemic.

In terms of unwillingness to work during the pandemic, our study revealed that more than one in three FHWs (35.8%) reported their willingness, which is a considerable issue because the health system’s workload during the pandemic is usually presumed to higher, and all available health resources are required to combat emergencies. The unwillingness rate to work during the pandemic observed in our study was higher than those reported in several other studies among health workers during public health emergencies [12,13,14,32,33]. In the present study, these high rates may be due to inadequate knowledge (17%) and pre-existing chronic disease (13.5%), and shortage of PPE (86%) (data not presented in the table), and other factors such as unknown risks, and emergency preparedness competencies [33,34]. The high rates of unwillingness to work during the pandemic revealed by our study demand that additional efforts be made to rectify the situation. Interestingly, health workers by professional category, province, presence of chronic disease, dependent family members, and knowledge about COVID 19 were associated with a willingness to work during the pandemic. We observed that nurses, paramedics, public health workers, and laboratory staff were more willing to work than clinicians, which contradicts the results of a systematic review conducted by Aoyagi et al. [35], who reported healthcare workers’ willingness to work during an influenza pandemic was moderately high, albeit highly variable. We also found that FHWs from Karnali and Far-West provinces were more willing to work than their counterparts from Bagmati Province. These provincial differences could be due to virtually no cases of COVID-19 during the study, and that the FHWs were willing to work in a humane way. Furthermore, FHWs with adequate knowledge about COVID-19 were more willing to work, which concurs with a study performed on the 2007 influenza pandemic [36]. Our result shows that FHWs with a chronic disease and those that cared for family members were less willing to work, which is also in line with previous studies [13,35]. It may be caring for family members and coping with personal chronic health problems might diminish work-willingness.

We believe that the findings of this study could be meaningful to inform various stakeholders and policymakers at national and provincial level health departments who are involved in the drafting of future interventions to improve the effectiveness of the health sector during public health crises. However, despite our efforts, this study has some notable methodological limitations. First, data were obtained using a questionnaire over the web, and healthcare workers were enrolled using their networks. Therefore, our results should not be extended to healthcare workers who do not use the internet. Second, the data used were self-reported, which makes the study prone to desirability bias and inaccuracies. Third, participants were asked to consider their willingness to work under hypothetical conditions that did not exist when Nepal comparatively observed fewer cases and fatalities during the study period. Fourth, we used the five-point Likert scale for outcome assessment which might not be specifically used for assessing related knowledge-specific variables. We recommend studies of the impacts of FHWs’ knowledge, perception, and willingness to work on health sector efficiency in public health emergencies.

## 5. Conclusions

We concluded that Nepalese health workers have some gaps in knowledge related to COVID-19, the majority have a negative perception of the government’s COVID-19 response, and nearly one- third of them were unwilling to work during the COVID-19 pandemic. Our study also revealed that FHWs at local public health facilities, pharmacists and Ayurvedic FHWs, those with chronic diseases were less likely, and male FHWs were more likely to have adequate knowledge of COVID-19. In contrast, nurses/midwives, public health workers, FHWs from Karnali Province and Far-West Province, and those who had adequate knowledge of COVID-19 were found to have higher odds of having satisfactory perceptions towards government response to COVID-19 management. In addition, FHWs such as paramedics, public health workers, laboratory workers, FHWs from Karnali Province and Far-West Province, and those having perceived the government’s response to COVID-19 as satisfactory compared to their counterparts were found to have incremental willingness to work during the COVID-19 pandemic. Our study suggests that prompt actions are required to improve health workers’ knowledge of COVID-19, address negative perceptions to government responses, and motivate them through monetary, non-monetary, and other likely specific measures to provide effective and efficient healthcare services during the pandemic. Additional studies on the impacts of frontline health workers’ knowledge, perception, and willingness to work on health sector efficiency in the context of public health emergencies should be undertaken.

## Figures and Tables

**Table 2 healthcare-08-00554-t002:** Factors associated with knowledge of COVID-19.

Knowledge about COVID-19	Inadequate Knowledge	Adequate Knowledge	OR (Univariable)	OR (Multivariable)
	(*n* = 181)	(*n* = 870)	(95% CI; *p*-Value)	(95% CI; *p*-Value) ^φ^
Gender				
Female	72 (22.1)	254 (77.9)	-	-
Male	109 (15.0)	616 (85.0)	1.60 (1.15–2.23, *p* = 0.005) **	1.60 (1.02–2.47, *p* = 0.036) *
Professional Category				
Doctor	60 (16.2)	311 (83.8)	-	-
Paramedics	57 (18.5)	251 (81.5)	0.85 (0.57–1.27, *p* = 0.423)	1.06 (0.65–1.75, *p* = 0.809)
Nurse/Midwife	35 (20.2)	138 (79.8)	0.76 (0.48–1.22, *p* = 0.246)	1.21 (0.67–2.18, *p* = 0.537)
Public Health Workers	11 (9.0)	111 (91.0)	1.95 (1.02–4.03, *p* = 0.054)	1.65 (0.78–3.72, *p* = 0.203)
Lab Worker	8 (15.7)	43 (84.3)	1.04 (0.49–2.48, *p* = 0.929)	0.92 (0.42–2.26, *p* = 0.851)
Other (Pharmacist, Ayurveda HW, etc.)	10 (38.5)	16 (61.5)	0.31 (0.14–0.73, *p* = 0.006) **	0.33 (0.14–0.80, *p* = 0.012) *
Type of Health Facility				
Federal and Provincial managerial agencies	12 (9.0)	122 (91.0)	-	-
Teaching Hospital	38 (19.9)	153 (80.1)	0.40 (0.19–0.77, *p* = 0.009) **	0.51 (0.23–1.09, *p* = 0.090)
Public Hospital	38 (15.8)	202 (84.2)	0.52 (0.25–1.01, *p* = 0.064)	0.66 (0.30–1.37, *p* = 0.284)
Private Hospital	15 (11.4)	117 (88.6)	0.77 (0.34–1.70, *p* = 0.516)	0.95 (0.40–2.24, *p* = 0.915)
Local public health facilities	72 (24.7)	220 (75.3)	0.30 (0.15–0.56, *p* < 0.001) ***	0.35 (0.17–0.68, *p* = 0.003) **
Local-level managerial agencies		56 (90.3)	0.92 (0.34–2.75, *p* = 0.871)	0.96 (0.34–2.95, *p* = 0.936)
Presence of Chronic Disease				
No	148 (16.3)	761 (83.7)	-	-
Yes	33 (23.2)	109 (76.8)	0.64 (0.42–1.00, *p* = 0.042) *	0.58 (0.37–0.91, *p* = 0.015) *

^φ^ Odds Ratios were obtained by multivariate logistic regression adjusted for gender, professional categories, health facility types, and presence of chronic disease. * *p*-value < 0.05 at the 5% level of significance ** *p*-value < 0.01 at the 5% level of significance *** *p*-value < 0.001 at the 5% level of significance.

**Table 3 healthcare-08-00554-t003:** Factors associated with self-reported perception of government response to COVID-19 pandemic.

Self–Reported Perception of Government Response	Unsatisfactory Government Response	Satisfactory Government Response	OR (Univariable)	OR (Multivariable) ^φ^
	(*n* = 668)	(*n* = 383)		
Caste/ethnicity				
Brahmin/Chhetri	367 (60.9)	236 (39.1)	-	-
Madhesi/Muslim	145 (69.4)	64 (30.6)	0.69 (0.49–0.96, *p* = 0.028) *	1.15 (0.70–1.89, *p* = 0.586)
Janajati	115 (66.1)	59 (33.9)	0.80 (0.56–1.13, *p* = 0.211)	0.96 (0.65–1.42, *p* = 0.846)
Dalit	24 (70.6)	10 (29.4)	0.65 (0.29–1.34, *p* = 0.260)	0.67 (0.29–1.46, *p* = 0.332)
Sanyasi, Bharati	17 (54.8)	14 (45.2)	1.28 (0.61–2.64, *p* = 0.504)	1.13 (0.50–2.49, *p* = 0.770)
Professional Category				
Doctor	274 (73.9)	97 (26.1)	-	-
Paramedics	198 (64.3)	110 (35.7)	1.57 (1.13–2.18, *p* = 0.007)	1.18 (0.78–1.79, *p* = 0.439)
Nurse/Midwife	96 (55.5)	77 (44.5)	2.27 (1.55–3.31, *p* < 0.001) ***	2.10 (1.38–3.18, *p* < 0.001) ***
Public Health Workers	52 (42.6)	70 (57.4)	3.80 (2.49–5.85, *p* < 0.001) ***	1.83 (1.07–3.11, *p* = 0.027) *
Lab Worker	30 (58.8)	21 (41.2)	1.98 (1.07–3.60, *p* = 0.027)	1.52 (0.79–2.90, *p* = 0.207)
Other (Pharmacist, Ayurveda HW etc.)	18 (69.2)	8 (30.8)	1.26 (0.50–2.89, *p* = 0.606)	1.37 (0.52–3.38, *p* = 0.506)
Province				
Bagmati Province	147 (72.1)	57 (27.9)	-	-
Province 1	136 (70.5)	57 (29.5)	1.08 (0.70–1.67, *p* = 0.726)	0.99 (0.62–1.59, *p* = 0.976)
Province 2	113 (72.4)	43 (27.6)	0.98 (0.61–1.56, *p* = 0.937)	0.88 (0.48–1.61, *p* = 0.680)
Gandaki Province	42 (60.9)	27 (39.1)	1.66 (0.93–2.93, *p* = 0.083)	1.69 (0.92–3.11, *p* = 0.090)
Province 5	96 (62.7)	57 (37.3)	1.53 (0.98–2.40, *p* = 0.062)	1.48 (0.92–2.40, *p* = 0.105)
Karnali Province	46 (41.4)	65 (58.6)	3.64 (2.25–5.96, *p* < 0.001) ***	2.62 (1.52–4.53, *p* = 0.001) **
Far–West Province	88 (53.3)	77 (46.7)	2.26 (1.47–3.49, *p* < 0.001) ***	1.72 (1.06–2.80, *p* = 0.030) *
Type of Health Facility				
Federal and Provincial managerial agencies	53 (39.6)	81 (60.4)	-	-
Teaching Hospitals	135 (70.7)	56 (29.3)	0.27 (0.17–0.43, *p* < 0.001) ***	0.52 (0.29–0.93, *p* = 0.027) *
Public Hospitals	168 (70.0)	72 (30.0)	0.28 (0.18–0.43, *p* < 0.001) ***	0.41 (0.24–0.70, *p* = 0.001) **
Private Hospitals	90 (68.2)	42 (31.8)	0.31 (0.18–0.50, *p* < 0.001) ***	0.52 (0.28–0.94, *p* = 0.032)
Local public health facilities	196 (67.1)	96 (32.9)	0.32 (0.21–0.49, *p* < 0.001) ***	0.49 (0.30–0.81, *p* = 0.005) **
Local-level managerial agencies	26 (41.9)	36 (58.1)	0.91 (0.49–1.68, *p* = 0.752)	1.12 (0.58–2.20, *p* = 0.742)
Knowledge about COVID–19				
Inadequate	154 (85.1)	27 (14.9)	-	-
Adequate	514 (59.1)	356 (40.9)	3.95 (2.61–6.20, *p* < 0.001) ***	3.86 (2.51–6.16, *p* < 0.001) ***

^φ^ Odds Ratios were obtained by multivariate logistic regression adjusted for caste/ethnicity, Professional category, Province, type of health facility, and health worker perceived knowledge of COVID-19 * *p*-value < 0.05 at the 5% level of significance ** *p*-value < 0.01 at the 5% level of significance *** *p*-value < 0.001 at the 5% level of significance.

**Table 4 healthcare-08-00554-t004:** Factors associated with self-reported willingness to work during the COVID-19 pandemic.

Perceived Willingness to Work	Unwilling to Work	Willing to Work	OR (Univariable)	OR (Multivariable)
	(*n* = 377)	(*n* = 674)	(95% CI, *p*–Value)	(95% CI, *p*–Value) ^φ^
Caste/ethnicity				
Brahmin/Chhetri	203 (33.7)	400 (66.3)	-	-
Madhesi/Muslim	89 (42.6)	120 (57.4)	0.68 (0.50–0.95, *p* = 0.021) *	1.11 (0.69–1.80, *p* = 0.659)
Janajati	69 (39.7)	105 (60.3)	0.77 (0.55–1.10, *p* = 0.145)	0.90 (0.62–1.31, *p* = 0.575)
Dalit	7 (20.6)	27 (79.4)	1.96 (0.88–4.95, *p* = 0.121)	1.74 (0.76–4.53, *p* = 0.215)
Other (Sanyasi, Giri, etc.)	9 (29.0)	22 (71.0)	1.24 (0.58–2.89, *p* = 0.595)	0.97 (0.43–2.36, *p* = 0.945)
Professional Category				
Doctor	191 (51.5)	180 (48.5)	-	-
Paramedics	83 (26.9)	225 (73.1)	2.88 (2.09–3.99, *p* < 0.001) ***	2.52 (1.79–3.58, *p* < 0.001) ***
Nurse/Midwife	52 (30.1)	121 (69.9)	2.47 (1.69–3.64, *p* < 0.001) ***	2.09 (1.40–3.17, *p* < 0.001) ***
Public Health Workers	28 (23.0)	94 (77.0)	3.56 (2.26–5.77, *p* < 0.001) ***	2.40 (1.47–4.01, *p* = 0.001) **
Lab Worker	11 (21.6)	40 (78.4)	3.86 (1.98–8.11, *p* < 0.001) ***	3.54 (1.77–7.61, *p* = 0.001) **
Other (Pharmacist, Ayurveda HW etc.)	12 (46.2)	14 (53.8)	1.24 (0.56–2.79, *p* = 0.600)	1.24 (0.54–2.89, *p* = 0.609)
Province				
Bagmati Province	90 (44.1)	114 (55.9)	-	-
Province 1	72 (37.3)	121 (62.7)	1.33 (0.89–1.99, *p* = 0.168)	1.18 (0.77–1.81, *p* = 0.446)
Province 2	71 (45.5)	85 (54.5)	0.95 (0.62–1.44, *p* = 0.792)	0.83 (0.47–1.45, *p* = 0.510)
Gandaki Province	29 (42.0)	40 (58.0)	1.09 (0.63–1.90, *p* = 0.762)	1.24 (0.69–2.22, *p* = 0.473)
Province 5	64 (41.8)	89 (58.2)	1.10 (0.72–1.68, *p* = 0.666)	0.87 (0.56–1.37, *p* = 0.554)
Karnali Province	17 (15.3)	94 (84.7)	4.37 (2.48–8.06, *p* < 0.001) ***	2.96 (1.62–5.64, *p* = 0.001) **
Far–West Province	34 (20.6)	131 (79.4)	3.04 (1.92–4.90, *p* < 0.001) ***	2.10 (1.28–3.48, *p* = 0.004) **
Presence of Chronic Disease				
No	313 (34.4)	596 (65.6)	-	-
Yes	64 (45.1)	78 (54.9)	0.64 (0.45–0.92, *p* = 0.015) *	0.67 (0.46–0.99, *p* = 0.043) *
HWs having family members who need care				
No	120 (31.5)	261 (68.5)	-	-
Yes	257 (38.4)	413 (61.6)	0.74 (0.57–0.96, *p* = 0.026) *	0.72 (0.54–0.95, *p* = 0.021) *
Perceived Knowledge about COVID–19				
Inadequate	86 (47.5)	95 (52.5)	-	-
Adequate	291 (33.4)	579 (66.6)	1.80 (1.30–2.49, *p* < 0.001) ***	1.81 (1.27–2.58, *p* = 0.001) **
Perception of government response				
Unsatisfactory perception	264 (39.5)	404 (60.5)	-	-
Satisfactory Perception	113 (29.5)	270 (70.5)	1.56 (1.20–2.05, *p* = 0.001) **	1.12 (0.83–1.51, *p* = 0.448)

**^φ^** Odds Ratios were obtained by multivariate logistic regression adjusted for caste/ethnicity, Professional category, Province, Presence of Chronic Disease, health workers with family members requiring care, perceived knowledge of COVID-19, and perception of government response. * *p*-value < 0.05 at the 5% level of significance ** *p*-value < 0.01 at the 5% level of significance *** *p*-value < 0.001 at the 5% level of significance.

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
