# Peer review of "Frontline Healthcare Workers’ Knowledge and Perception of COVID-19, and Willingness to Work during the Pandemic in Nepal"

_healthcare, 2020, doi:10.3390/healthcare8040554_

Round 1

Reviewer 1 Report

The authors have investigated some of the key factors that are associated with frontline healthcare works during COVID-19 in Nepal. This work should have been published earlier, however, it important to highlight the knowledge of healthcare workers about any pandemic before serving as frontline health workers.

Overall, the manuscript is well written. However, would suggest the authors thoroughly proofread the manuscript. There are serval grammatical mistakes and redundancy. Also, the author should limit the use of personal pronouns in the manuscript-it is not bad; but, too many in a sentence or paragraph makes it look untidy. Avoid repetition of words throughout the text. E.g.,

Line 92: The present study describes the actual HW scenarios factors associated with their knowledge of 93 COVID-19, their reactions to government interventions, and, most importantly, their perceived 94 willingness to work during the pandemic.

This study might also provide valuable and actionable information to Nepalese policymakers to allow the judicious allocation of scarce resources in the short run.

Revise the following sentences:

-As an initial step, we first appointed a doctor or public health professional in each 109 of the seven provinces to act as a coordinator and co-investigator in the team.

-These seven coordinators then sent HWs known to them a link to our questionnaire and asked that these individuals send a Google link to other HWs they knew.

-The questionnaire took approximately 10 minutes to complete.

-In the present study, FHWs were defined as doctors, nurses, paramedics, laboratory workers, pharmacists, pathologist, technical personnel, public health workers, and others directly involved in COVID-19 prevention and treatment that have direct contact with confirmed or suspected cases during patient intake, screening, inspection, testing, transport, treatment, nursing, specimen collection, or pathogen detection.

Our observation demonstrated that nearly two-thirds FHWs (63.6%) believed the government response to COVID-19 was unsatisfactory

Cite supporting reference  and change “diseases” to” infection”

Based on available evidence, the diseases is transmitted between individuals via nasopharyngeal droplets or saliva.

Table 4. I wonder what has religion and Ethnicity to do here. I just Google it and it seems all these names are based on the caste system. It would have been great if it was done based on age or area rather than cast. Because it shows which group is willing to work in a high number and this data might create differences based on the cast. I think there is already data based on the province. So, data based on cast should be removed.

Cite more supporting references. Besides, I would suggest the authors provide the questionnaire as a supporting file.

Author Response

REVIEWER 1: ROUND FIRST

Journal: Healthcare

Manuscript ID: healthcare-1010740

Title: Frontline healthcare workers' knowledge and perception of COVID-19, and willingness to work during the pandemic in Nepal

Overall authors’ response:

Thank you very much for your fruitful comments and suggestions, and we greatly appreciate reviewer’s comments and suggestions. We have extensively revised the manuscript based on reviewers’ comments and suggestions. All modifications are marked with blue colored writing in revised version of the manuscript to allow reviewers’ verifications. We have provided the point to point clarification of the comments in the manuscript as suggested.

Comments and Suggestions for Authors, and authors’ response

The authors have investigated some of the key factors that are associated with frontline healthcare works during COVID-19 in Nepal. This work should have been published earlier, however, it is important to highlight the knowledge of healthcare workers about any pandemic before serving as frontline health workers.

Overall, the manuscript is well written. However, would suggest the authors thoroughly proofread the manuscript. There are serval grammatical mistakes and redundancy. Also, the author should limit the use of personal pronouns in the manuscript-it is not bad; but, too many in a sentence or paragraph makes it look untidy. Avoid repetition of words throughout the text. E.g.,

Comments: Line 92: The present study describes the actual HW scenarios factors associated with their knowledge of 93 COVID-19, their reactions to government interventions, and, most importantly, their perceived 94 willingness to work during the pandemic.

This study might also provide valuable and actionable information to Nepalese policymakers to allow the judicious allocation of scarce resources in the short run.

Response: Agree. Revised.

Revise the following sentences:

Comment: -As an initial step, we first appointed a doctor or public health professional in each of the seven provinces to act as a coordinator and co-investigator in the team.

Response: Agree. Revised.

Comments: -These seven coordinators then sent HWs known to them a link to our questionnaire and asked that these individuals send a Google link to other HWs they knew.

-The questionnaire took approximately 10 minutes to complete.

Response: Agree. Corrected.

Comments: In the present study, FHWs were defined as doctors, nurses, paramedics, laboratory workers, pharmacists, pathologist, technical personnel, public health workers, and others directly involved in COVID-19 prevention and treatment that have direct contact with confirmed or suspected cases during patient intake, screening, inspection, testing, transport, treatment, nursing, specimen collection, or pathogen detection.

 Our observation demonstrated that nearly two-thirds FHWs (63.6%) believed the government response to COVID-19 was unsatisfactory

Response: Agree. Revised.

Comments: Cite supporting reference and change “diseases” to” infection”

Based on available evidence, the diseases is transmitted between individuals via nasopharyngeal droplets or saliva.

Response: Agree. Revised with appropriate citation as suggested.

Comments: Table 4. I wonder what has religion and Ethnicity to do here. I just Google it and it seems all these names are based on the caste system. It would have been great if it was done based on age or area rather than cast. Because it shows which group is willing to work in a high number and this data might create differences based on the cast. I think there is already data based on the province. So, data based on cast should be removed. Cite more supporting references. Besides, I would suggest the authors provide the questionnaire as a supporting file.

Response: Strongly agree. We have revised the term ethnicity to caste/ethnicity as an appropriate word and also clearly defined the outcome measurements in the methods section of the revised manuscript.

Reviewer 2 Report

Upadhyaya et al .: "Frontline healthcare workers’ knowledge and perception of COVID-19 ... in Nepal "

The manuscript examines the knowledge of health professionals about COVID-19 and relates it to the willingness to work on the front lines with the sick. The introduction describes well the outbreak of the pandemic.

In chapter 2.2 (Survey instrument and data collection) I am missing a clever reason why the limit was set at a cut-off of 60%? Instead of making a rough dichotomous distinction between “good” and “bad” knowledge, one could in principle have introduced a category of “medium knowledge”. It might have been even better to dispense with a classification and - as far as possible - to calculate correlations (e.g. between years of professional experience and knowledge about COVID-19).

Which questions were asked to test the knowledge about Covid-19, some examples would be helpful for the reader. E.g. under "knowledge of the causative agent ..." I can't imagine anything, what were the possible answers?

Even under “Government response” one cannot imagine anything like that. What exactly do the authors mean here, what questions were asked?

A 5-point Likert scale does not seem to me to be the right instrument here. Questions about specific knowledge of a disease should give a specific answer, but not a 5-point answer scale (as said in the upper part of this paragraph).

Were there any inclusion and exclusion criteria for the participants? Were health care workers in rural areas also reached (internet connection)?

Which test was used for the “univariate analysis”? Were the data normally distributed and did the data show homogeneity of variance? Were parametric or non-parametric calculation methods used for the evaluation?

In regression analysis, one variable predicts the outcome of another variable. If you don't know which variable is causing the other, correlation coefficients would be a better option.

In Chapter 3 (Results), the standard deviation, minimum and maximum are missing for age. Otherwise the sample is well described.

The authors write that men had better knowledge about COVID-19. Have the authors checked whether e.g. there were more male doctors and more female nurses, lab-workers etc.  in your sample? Doctors probably have better knowledge than other professional groups (nurses and e.g. laboratory workers are also often female). These data are only conclusive if they are put in relation to the profession.

I also miss an age comparison here: is there a significant correlation between years of professional experience and knowledge of Covid-19?

The authors also compared Public Health Facilities and Federal and Provincial Manegerial Agencies. Here, too, the question is which professional training (doctor, nurse, laboratory worker, pharmacist, etc.) and what number of years of professional experience do the members of the individual institutions have? Conclusions can only be drawn if these groups are comparable.

The analysis by provinces of Nepal could be interesting if the reader would know more about these provinces (number of inhabitants, location, many large cities or rather rural, industry or agriculture?).

Some graphs of the results could be helpful.

The discussion is well written and compares the results with some other studies.

Author Response

REVIEWER 2: ROUND FIRST

Journal: Healthcare

Manuscript ID: healthcare-1010740

Title: Frontline healthcare workers' knowledge and perception of COVID-19, and willingness to work during the pandemic in Nepal

Overall authors’ response:

Thank you very much for your fruitful comments and suggestions, and we greatly appreciate the reviewer’s comments and suggestions. We have extensively revised the manuscript based on reviewers’ comments and suggestions. All modifications are marked with blue colored writing in the revised version of the manuscript to allow reviewers’ verifications. We have provided the point to point clarification of the comments in the manuscript as suggested.

Comments and Suggestions for Authors, and authors’ response

Comment: Upadhyaya et al .: "Frontline healthcare workers’ knowledge and perception of COVID-19 ... in Nepal "

The manuscript examines the knowledge of health professionals about COVID-19 and relates it to the willingness to work on the front lines with the sick. The introduction describes well the outbreak of the pandemic.

Comment: In chapter 2.2 (Survey instrument and data collection) I am missing a clever reason why the limit was set at a cut-off of 60%? Instead of making a rough dichotomous distinction between “good” and “bad” knowledge, one could in principle have introduced a category of “medium knowledge”. It might have been even better to dispense with classification and - as far as possible - to calculate correlations (e.g. between years of professional experience and knowledge about COVID-19).

Response: Partially agree. As per the expert suggestions and opinion, we came to categorize it into only two as the COVID-19 pandemic information seems important for immediate intervention if HWs do have a cut-off score of 60%.

Comments: Which questions were asked to test the knowledge about Covid-19, some examples would be helpful for the reader. E.g. under "knowledge of the causative agent ..." I can't imagine anything, what were the possible answers? Even under “Government response” one cannot imagine anything like that. What exactly do the authors mean here, what questions were asked? A 5-point Likert scale does not seem to me to be the right instrument here. Questions about specific knowledge of disease should give a specific answer, but not a 5-point answer scale (as said in the upper part of this paragraph). Were there any inclusion and exclusion criteria for the participants? Were health care workers in rural areas also reached (internet connection)?

Response: Agree. Thank you very much for this excellent comment. We have now extensively revised the methods section to make it easily understandable including the exclusion criteria. Besides, we have clarified the things wherever required. We also addressed the issue of the use of a 5-point Likert scale for knowledge assessment as one of the limitations of the study.

Comment: Which test was used for the “univariate analysis”? Were the data normally distributed and did the data show homogeneity of variance? Were parametric or non-parametric calculation methods used for the evaluation? In regression analysis, one variable predicts the outcome of another variable. If you don't know which variable is causing the other, correlation coefficients would be a better option. In Chapter 3 (Results), the standard deviation, minimum, and maximum are missing for age. Otherwise, the sample is well described.

Response: For your kind information, we used the Univariate logistic regression analysis (enter method) for each variable to identify the one to one association between predictor and outcome variable. Multivariable models were created for each dependent variable and included independent variables found to be significant (p-value <0.05) by univariable analysis. The data did not show the overly inflated outliers. We also checked for multicollinearity before entering variables to multivariable logistic regression analysis. Coefficients in the regression models were transformed into odds ratios with 95 % confidence intervals. P-values of <0.05 were considered significant. Further, we have mentioned IQR for age, for your kind information.

Comment: The authors write that men had better knowledge of COVID-19. Have the authors checked whether e.g. there were more male doctors and more female nurses, lab-workers, etc. in your sample? Doctors probably have better knowledge than other professional groups (nurses and e.g. laboratory workers are also often female). These data are only conclusive if they are put in relation to the profession. I also miss an age comparison here: is there a significant correlation between years of professional experience and knowledge of Covid-19? The authors also compared Public Health Facilities and Federal and Provincial Managerial Agencies. Here, too, the question is which professional training (doctor, nurse, laboratory worker, pharmacist, etc.) and what number of years of professional experience do the members of the individual institutions have? Conclusions can only be drawn if these groups are comparable. The analysis by provinces of Nepal could be interesting if the reader would know more about these provinces (number of inhabitants, location, many large cities or rather rural, industry or agriculture?). Some graphs of the results could be helpful. The discussion is well written and compares the results with some other studies.

Response: Agree. Thank you very much for this comment. We have discussed the issue clearly about the knowledge differences among different FHWs including doctors, besides, having discussed the clues on provincial differences. Our investigators’ team reached to the consensus that COVID-19 is a newly emerged disease to every FHWs and many pieces of stuff are still not known to FHWs. Therefore, we did not report it by work experience although it could be of interest to the readers.

Reviewer 3 Report

General comments

The present paper is an original article that investigated the contextual factors associated with the knowledge, perceptions, and willingness of frontline healthcare workers to work during the COVID-19 pandemic in Nepal.

First of all, I'm surprised by the number of authors, specifically 11, which developed and pretest the questionnaire. Collaboration with researchers from the USA and Japan, and of different disciplines. I think it is important to highlight this in order to assess the quality of the questionnaire. Which I see as a positive point.

Specific comments

  1. Title: it is too long. A total of 24 words. Please, a title must be clearer and shorter.
  2. Abstract: it must be structured following the instructions of the journal, please modify it. It's better to understand as well. Also, it seems to be more than 200 words permitted.
  3. Introduction: paragraphs are too long. Please divide it into shorter paragraphs.
  4. Some corrections in the introduction: line 70, fatalities refer to deaths? Line the word Nepalese should be in small letters.
  5. Line 104: capital letter after a full stop.
  6. What are the criteria to include or exclude the participants? Why participants between 18 to 60?
  7. Page 6: most of the page is blank. You must adjust that. I know it's because of the following table, but everything must be more uniform. 
  8. Table 2: adjust until it is on 1 page only as it looks very bad as they have left it.
  9. You have to add the page number to the next pages from table 2 to the last table.
  10. The tables are too big, please reduce them.
  11. Discussion: it is the first study in Nepal but there are other studies in this topic.
  12. Discussion: paragraphs are too long.
  13. Line 308 Nepalese should be in small letters.
  14. The format of "authors contributions" to "conflict of interests" must be in the same format as the manuscript. Please, review.
  15. Please review the references. There are not correct.

Final comments

In my opinion, the work is very well done, the content is good, the results and conclusions are good. However, the formatting would be improved, the paragraphs would be smaller, there are small spelling mistakes and written English could be improved. I congratulate the authors for the work and encourage them to make these changes in order to accept this article as a publication in Healthcare. Thank you for considering the journal for publication and continue to make studies as relevant as this one.

Author Response

REVIEWER 3: ROUND FIRST

Journal: Healthcare

Manuscript ID: healthcare-1010740

Title: Frontline healthcare workers' knowledge and perception of COVID-19, and willingness to work during the pandemic in Nepal

Overall authors’ response:

Thank you very much for your fruitful comments and suggestions, and we greatly appreciate reviewer’s comments and suggestions. We have extensively revised the manuscript based on reviewers’ comments and suggestions. All modifications are marked with blue colored writing in revised version of the manuscript to allow reviewers’ verifications. We have provided the point to point clarification of the comments in the manuscript as suggested.

Comments and Suggestions for Authors, and authors’ response

General comments

The present paper is an original article that investigated the contextual factors associated with the knowledge, perceptions, and willingness of frontline healthcare workers to work during the COVID-19 pandemic in Nepal. First of all, I'm surprised by the number of authors, specifically 11, which developed and pretest the questionnaire. Collaboration with researchers from the USA and Japan, and of different disciplines. I think it is important to highlight this in order to assess the quality of the questionnaire. Which I see as a positive point.

Specific comments

  1. Title: it is too long. A total of 24 words. Please, a title must be clearer and shorter.

Response: Agree. Title shortened as suggested.

  1. Abstract: it must be structured following the instructions of the journal, please modify it. It's better to understand as well. Also, it seems to be more than 200 words permitted.

Response: Agree. Revised

  1. Introduction: paragraphs are too long. Please divide it into shorter paragraphs.

 Response: Agree. Corrected

  1. Some corrections in the introduction: line 70, fatalities refer to deaths? Line the word Nepalese should be in small letters.

Response: Agree. Corrected. But the term ‘Nepalese’ is correct.

  1. Line 104: capital letter after a full stop.

Response: Agree. Corrected

  1. What are the criteria to include or exclude the participants? Why participants between 18 to 60?

Response: Agree. Exclusion criteria mentioned. The health workers in Nepal get retired after age of 60 years and 18th year is the minimum entry required years.

  1. Page 6: most of the page is blank. You must adjust that. I know it's because of the following table, but everything must be more uniform. 

Response: Agree. Corrected

  1. Table 2: adjust until it is on 1 page only as it looks very bad as they have left it.

Response: Agree. Corrected

  1. You have to add the page number to the next pages from table 2 to the last table.

Response: Agree. Corrected

  1. The tables are too big, please reduce them.

Response: Agree. Corrected

  1. Discussion: it is the first study in Nepal but there are other studies in this topic.

Response: Agree. Revised.

  1. Discussion: paragraphs are too long.

Response: Agree. Corrected.

  1. Line 308 Nepalese should be in small letters.

Response: Disagree. It’s fine.

The format of "authors contributions" to "conflict of interests" must be in the same format as the manuscript. Please, review.

Response: Agree. Revised

Please review the references. There are not correct.

Response: Agree. Revised.

Final comments

In my opinion, the work is very well done, the content is good, the results and conclusions are good. However, the formatting would be improved, the paragraphs would be smaller, there are small spelling mistakes and written English could be improved. I congratulate the authors for the work and encourage them to make these changes in order to accept this article as a publication in Healthcare. Thank you for considering the journal for publication and continue to make studies as relevant as this one.

Response: Agree. We thank you very much for your great comments and suggestions as a result of which our manuscript have been improved extensively.